# Role of microRNAs in Pressure Ulcer Immune Response, Pathogenesis, and Treatment

**DOI:** 10.3390/ijms22010064

**Published:** 2020-12-23

**Authors:** Stephen M. Niemiec, Amanda E. Louiselle, Kenneth W. Liechty, Carlos Zgheib

**Affiliations:** 1Laboratory for Fetal and Regenerative Biology, Department of Surgery, University of Colorado Denver School of Medicine, Aurora, CO 80045, USA; Stephen.niemiec@cuanschutz.edu (S.M.N.); Amanda.louiselle@cuanschutz.edu (A.E.L.); Ken.liechty@cuanschutz.edu (K.W.L.); 2Division of Pediatric Surgery, Children’s Hospital Colorado, Aurora, CO 80045, USA

**Keywords:** pressure ulcer, microRNA (miRNA, miR), chronic wound

## Abstract

Pressure ulcers are preventable, yet highly prevalent, chronic wounds that have significant patient morbidity and high healthcare costs. Like other chronic wounds, they are characterized by impaired wound healing due to dysregulated immune processes. This review will highlight key biochemical pathways in the pathogenesis of pressure injury and how this signaling leads to impaired wound healing. This review is the first to comprehensively describe the current literature on microRNA (miRNA, miR) regulation of pressure ulcer pathophysiology.

## 1. Introduction

Pressure ulcers are chronic wounds associated with significant morbidity and are a substantial burden to the healthcare system. Pressure wounds are considered a preventable disease, and the National Quality Forum identified the development of a severe pressure ulcer as a Serious Reportable Event that will not receive Centers for Medicaid and Medicare Services (CMS) reimbursement [1]. Despite this, pressure injury affects 6–18% of hospitalized patients [2,3], with significantly higher prevalence in chronically ill or bedridden individuals, such as patients in the intensive care unit or those with spinal cord injury [4,5,6]. Patients who develop pressure ulcers have significant psychosocial burden associated with the disease, including increased anxiety and feelings of social isolation, in addition to the physical morbidity associated with prolonged hospital stays and increased risk of infection [7]. The average hospital length of stay for patients diagnosed with a pressure ulcer is three times that of patients who do not develop pressure ulcers, inciting additional care-related costs of USD 8200–25,000 per stay and over USD 11 billion annually [5,8,9,10].

## 2. Pathogenesis of Impaired Healing in Pressure Ulcers

Pressure ulcers, also known as bed sores and decubitus ulcers, are skin and soft tissue wounds that develop due to pressure injury. Most often seen at sites overlying bony prominences, pressure injury occurs when external forces on the skin and soft tissue ultimately result in internal damage and skin ulceration; however, the physiologic derangements leading to wound formation are complex and multifactorial [5,7]. Patient populations with decreased mobility are at particular risk of pressure injury due to prolonged mechanical force during periods of static positioning, including patients with spinal cord injury as well as patients undergoing prolonged operations [4,5,6,11]. 

Pressure ulcer severity is most often defined using the National Pressure Injury Advisory Panel (NPIAP) staging system, although other classification systems are also accepted [7,12,13,14]. The revised NPIAP staging system classifies wounds into six degrees of severity—five determined by the depth of wound invasion into the underlying soft tissue and one unstageable classification. Severe pressure ulcers can progress to deep soft tissue wounds, with risk of developing osteomyelitis if not treated promptly. As a pressure ulcer heals, it maintains its worst historical stage, as more severe wounds are prone to delayed healing [15]. 

The study of pressure ulcer pathophysiology in preclinical and clinical models is challenging, so correlations with other models of chronic wound healing are often used to guide future investigation [16]. However, the mechanism of pressure injury has unique elements that require more specific preventative and therapeutic intervention. Major components of pressure ulcer pathogenesis include ischemia-reperfusion injury, poor lymphatic drainage, cellular deformation, and excess cellular apoptosis and extracellular matrix (ECM) breakdown resulting in a chronic inflammatory state and dysregulated healing [17,18]. Immunocompromised patients, such as those with diabetes or chronically ill patients in the ICU, are particularly prone to the development of chronic pressure ulcers due to immune system dysregulation and impaired wound healing [19]. The pathways described below that are involved in the pathogenesis of pressure ulceration are depicted in Figure 1.

### 2.1. Ischemia-Reperfusion Injury and Dysregulated Immune Response

Ischemia-reperfusion events within chronic pressure ulcer pathogenesis typically follow a repetitive pattern of insults, rather than a single episode of ischemia followed by reperfusion [20]. Mechanical pressure causes capillary and venous obstruction that results in local tissue ischemia and edema, often starting in deep tissues and extending superficially [17]. The process of ischemia results in reduced ATP generation and impaired mitochondrial oxidative phosphorylation, increased complement and leukocyte activation, and elevated levels of inflammation [20]. Hypoxic conditions in pressure ulcers result in increased cell death and release of toxic metabolites such as cyclooxygenase (COX)-2 and interleukin (IL)-6 through upregulation of extracellular signal-regulated kinase (Erk) and p38 mitogen-activated protein kinase (MAPK) [21,22,23,24,25]. Meanwhile, venous and lymphatic obstruction results in a decrease in metabolite clearance, promoting a sustained inflammatory response [26]. Toxic metabolites such as COX-2 and IL-6 stimulate neutrophils and macrophages to release more pro-inflammatory cytokines such as tissue necrosis factor alpha (TNFα) and IL-8, which perpetuates the inflammatory cycle [18]. 

When pressure is alleviated, restoration of blood flow leads to an increase in reactive oxygen species (ROS) and oxidative stress. Oxidative stress is known to promote an M1, pro-inflammatory, macrophage phenotype in chronic wounds such as pressure ulcers and diabetic wounds, resulting in sustained inflammatory signaling and impaired wound healing [18,27]. ROS further inhibit the phosphatidylinositol 3-hydroxy kinase/protein kinase B (PI3K/AKT) pathway, an anti-apoptotic pathway that has protective effects on ischemia-reperfusion injury but is downregulated during pressure ulcer development [28,29]. During reperfusion, there is additionally a relative decrease in nitric oxide expression, leading to impaired vasorelaxation and poor clearance of apoptotic cells and debris. The lack of vascular relaxation leads to increases in edema and leukocyte trapping, which exacerbates ROS and pro-inflammatory signal production during future ischemia-reperfusion cycles [30,31].

### 2.2. Impaired Lymphatic Drainage

Mechanical deformation leads to compression of blood vessels, but also to the sustained collapse of lymphatic channels [32,33,34]. The lymphatic system works to absorb interstitial fluid, clear toxins, and traffic immune cells from soft tissues into the arteriovenous circulation. Increasing skin load has been shown to decrease lymphatic clearance in pressure ulcer models [35]. Lymphatic vessels function to clear toxic metabolites and excess fluid from the interstitial space, and when obstructed, this lack of debris clearance results in increased inflammation and local cell death [26]. Lymphedema, or the build-up of lymphatic fluid in the interstitial space, is a component of pressure ulcer pathophysiology, but is often studied as a unique entity that can induce ulcer formation. Acquired lymphedema results in increased levels of pro-inflammatory molecules such as TNFα, IL-6, IL-8, and monocyte chemoattractant protein-1 (MCP-1), which contribute to an increase in the infiltration of pro-inflammatory cells, as well as a decrease in T-regulatory cells involved in the resolution of inflammation [36,37]. The impairment of lymphatic clearance, along with impaired venous drainage described above, leads to the maintenance of high leukocyte infiltrate with poor ROS and cytokine clearance. 

### 2.3. Cellular Deformation and Apoptosis

External force results in cellular deformation, and the mechanical signaling can induce cellular-level and tissue-level distortion that impairs cell viability [38]. Cellular damage occurs both when a high load is applied for short periods of time and when small loads are applied for extended periods of time [39]. Cellular deformation can lead to impairments in plasma membrane permeability and abnormal oxygen clearance in myocytes and soft tissues [40,41]. Sustained cellular deformation, as seen in pressure ulcers, leads to cytoskeletal damage and cell death [42,43]. 

Myocyte deformation can induce muscle tissue to release TNF-α, and with impairments in biowaste clearance due to lymphatic obstruction and impaired efferocytosis, there is secondary damage to surrounding cells [44]. TNFα can reduce protein synthesis within myocytes and activate caspase activity, an important inducer of apoptosis [45,46]. When myocytes are exposed to prolonged low stress compression, there is a build-up of toxic metabolites in the surrounding fluid, and if healthy myocytes are exposed to this biowaste, there is increased cellular damage [44].

Mechanical compression can also lead to increased plasma membrane permeability and an influx of calcium cations, resulting in mitochondrial activation and the generation of ROS [38]. Models of chronic compression have shown decreased PI3K/AKT activity, increased apoptosis, and elevated ROS [47,48]. Research in normal wound healing has shown that macrophage phagocytosis of apoptotic cells can induce anti-inflammatory pathways to promote wound healing. Apoptotic cells release chemotactic signaling molecules such as lysophosphatidylcholine (LPC) to attract macrophages to the site of injury; however, when not adequately cleared, these immune cells produce chronic inflammation [49,50]. Chronic wounds, such as diabetic wounds, have decreased PI3K/AKT activity with excessive endothelial and keratinocyte apoptosis while also displaying impaired clearance of apoptotic debris due to immune system dysregulation [51,52,53,54]. Pressure ulcers show a similar downregulation in PI3K/AKT [28].

### 2.4. Chronic Inflammation and the Dysregulated Immune Response in Pressure Ulcers

Normal wound healing follows a three-phase immune response with an initial inflammatory phase followed by proliferative and remodeling phases. Due to sustained injury mechanisms explained above, pressure ulcers remain in the inflammatory phase and are characterized by impaired healing. Mouse models of pressure ulcers have shown increased expression of MCP-1 and increased inflammatory cell infiltration compared to MCP-1 knockout mice [55]. Dysregulated responses by specific immune cell lines have been implicated in chronic wound and pressure ulcer pathogenesis. Chronic pressure ulcers have excessive neutrophil infiltration within granulation tissue, and these cell lines produce elastase and collagenase that lead to ongoing extracellular matrix breakdown and impaired wound healing [56]. In normal wound healing, neutrophils typically undergo apoptosis and efferocytosis early following injury, but pressure ulcers see sustained neutrophil numbers in the wound base [56,57]. 

Ongoing cycles of ischemia–reperfusion and impaired metabolite clearance following cell death maintain this infiltrate in a pro-inflammatory immune state with elevated levels of pro-inflammatory macrophages. These pro-inflammatory macrophages produce ROS and inflammatory cytokines like IL-6 and TNFα [18]. They also secrete matrix metalloproteases (MMP) 2 and 9, which contribute to ongoing ECM breakdown and impaired tissue regeneration [58]. In diabetic wounds, elevations in IL-6 and TNFα further lowers dermal fibroblast proliferation through downregulation of PI3K/AKT activity, resulting in impaired collagen deposition and wound contracture, although this interaction is not fully elucidated in pressure ulcer pathophysiology [59].

## 3. Current Therapies in Pressure Ulcer Management

The management of pressure injury starts with prevention, as reductions in external forces that stimulate the pathophysiologic cascade described above are paramount to minimizing soft tissue pressure injury. Mechanisms to reduce load and prevent pressure injury have been described in detail elsewhere, but include regular repositioning and use of advanced support surfaces that improve pressure redistribution [7,60,61]. Regular offloading can help to reduce the time under pressure at a single site, but high mechanical loads such as those seen in patients with obesity or in tissue overlying bony prominences can still produce injury after relatively short periods of mechanical stress, and offloading can contribute to reperfusion injury [20,39]. Medical devices such as urinary catheters and cervical collars can also promote mechanical stress, and therefore are bioengineered to reduced pressure and shear in an effort to lower areas of concentrated mechanical stress and friction [62]. The use of silicone dressings and topical agents to prevent pressure ulcer development have some evidence in reducing overall mechanical stress on underlying tissue, but require further investigation [63,64]. 

Optimization of nutrition is critical for wound healing, and has also been implicated in the prevention of pressure wound formation [65]. For example, vitamin D deficiency has been shown to increase the risk of hospital-acquired pressure injury and the intake of high-protein nutritional supplements reduces pressure ulcer incidence in patients with terminal illnesses [66,67]. Among individuals who develop pressure injury, over 75% are malnourished, and the malnutrition can be secondary to undernutrition as well as obesity [68,69].

Despite the development of preventative strategies, pressure ulcers affect nearly 3 million persons annually in the United States alone, so the development of therapeutics targeting immune dysregulation to improve wound healing is of ongoing interest [70]. Mechanisms to improve healing such as negative pressure wound therapy and hyperbaric oxygen therapy have been used in pressure ulcers and other chronic wounds [70,71,72]. In chronic wounds, negative pressure wound therapy is able to lower pro-inflammatory signaling within the wound bed by decreasing IL-6, iNOS, TNFα, and JNK expression levels [73]. Surgical management with regular debridement or tissue reconstruction can be required for more severe, non-healing pressure wounds. Biologic dressings such as with decellularized skin or treatment with growth factors or topical agents such as honey have been suggested to improve healing in pressure ulcers [74]. One class of regulatory molecules investigated in other models of chronic wounds but not yet studied as a therapeutic target in pressure ulcers is microRNAs.

## 4. Targeting microRNAs to Enhance Pressure Ulcer Healing

MicroRNAs (miRNA, miR) are 19- to 25-nucleotide length non-coding RNAs that have recently been discovered to have critical roles in the pathogenesis, regulation, and treatment of chronic wounds [75,76,77,78]. Many of the pathophysiologic pathways in the development of chronic wounds are regulated by miRNAs, and the implications of miRNA dysregulation in pressure injury are of growing interest. MiRNAs act as post-transcriptional regulators of protein-coding genes, playing roles in cellular proliferation and apoptosis, extracellular matrix modulation, and the immune response [79,80]. Although miRNAs have been studied as therapeutic targets in chronic wounds, particularly in diabetic wounds, their role in pressure injury pathogenesis is not fully understood [78]. The current literature on miRNA function in pressure ulcers is summarized in Table 1 and depicted in Figure 1.

### 4.1. Dysregulated miRNAs in Pressure Ulcer Inflammatory Response

MicroRNAs have been shown to play significant roles in normal wound healing, and their dysregulation leads to delayed healing and contributes to chronic wound formation. Research in diabetic wound healing has highlighted many dysregulated microRNA responses in angiogenesis, immune modulation, extracellular matrix remodeling, and re-epithelialization during the wound healing process [76,77,89,90,91]. In pressure ulcers, sustained inflammation within the wound bed leads to an abnormal immune response and delayed healing. miR-21 and miR-885-3p have recently been found to have dysregulated inflammatory responses in pressure ulcer studies. 

MiR-21 is an anti-inflammatory miRNA with immune modulating activity during normal and dysregulated wound healing. It has been shown to have a role in regulating cell proliferation and apoptosis in diabetic retinopathy and nephropathy [92]. The effects of miR-21 on dysregulated immune response and healing in diabetic wounds have been well characterized [93,94,95]. miR-21 is upregulated during the early phases of diabetic wound healing, and over-expression of miR-21 leads to increased expression of the pro-inflammatory cytokines IL-1β, TNFα, inducible nitric oxide synthase (iNos), and IL-6 and to the promotion of a pro-inflammatory M1 macrophage phenotype [27]. 

Song et al. found similar inflammatory dysregulation in a pressure ulcer model of keratinocytes exposed to lipopolysaccharide (LPS), an inflammatory/infection injury model. LPS injury results in initial miR-21 suppression and high levels of pro-inflammatory markers, but upregulation of miR-21 through treatment with emodin decreases levels of IL-6, IL-1β, COX-2, and iNOS compared to those treated with emodin and a miR-21 inhibitor [81]. MiR-21 inactivates nuclear factor kappa B (NFκB) signaling, lowering these pro-inflammatory markers. One mechanism by which this may occur in pressure ulcers is through dysregulation of macrophage efferocytosis and polarization. In normal wounds, over-expression of miR-21 increases macrophage efferocytosis and transition to an anti-inflammatory phenotype, lowering TNFα expression and NFκB activity [96,97,98]. MiR-21 also results in activation of PI3K/AKT signaling through upstream phosphatase and tensin homolog (PTEN) modulation, and treatment with emodin to upregulate miR-21 levels lowered apoptosis and improved keratinocyte viability [81]. 

One mechanism to optimize the prevention and healing of pressure ulcers in a clinical setting is through nutritional modification, such as with vitamin D. Vitamin D is known to have immune modulation activity during wound healing, and patients with vitamin D deficiency are at increased risk of pressure ulcer formation and other chronic wounds, although a causal mechanism is still unclear [67,99]. Ji et al. found that patients who develop pressure ulcers have increased rates of a vitamin D receptor (VDR) polymorphism, Rs739837, that prevents the binding of miR-885-3p. miR-885-3p normally increases VDR activity and its downstream immune activity, and a loss of miR-885-3p binding to VDR significantly increases the risk of pressure ulcer formation in a hospital setting [82]. This polymorphism has been implicated in increasing the risk of severe asthma and dental caries due to its impairment in vitamin D-related immune response [100,101]. 

MiR-885-3p has been implicated in other immune response and anti-neoplastic pathways. In patients with type-1 diabetes, miR-885-3p is downregulated and there is upregulation of pro-inflammatory cytokine expression by peripheral blood mononuclear cells. MiR-885-3p overexpression can result in a decrease in NFκB activity within these cells through inhibition of toll-like receptor (TLR)-4 [102]. Vitamin D has been shown to lower hypoxia-induced activation of NFκB and ROS production in cerebral endothelial cells [103]. How miR-885-3p and the VDR may interact to modulate the immune response in pressure ulcer development and wound healing is still under investigation; however, it could potentially involve modulation of the NFκB signaling pathway.

### 4.2. MiRNA Response to Hypoxia in Pressure Ulcers

Two microRNAs, miR-449a and miR-135b, have been implicated in pressure injury pathophysiology utilizing in vitro models of hypoxia [83,84]. MiR-449a has largely been studied as a tumor suppressor with roles in inhibiting a variety of malignancies, including cervical cancer, prostate cancer, and osteosarcoma [104,105,106]. Yu et al. evaluated the effect of hypoxia on miR-449a levels within HaCaT cells, immortalized keratinocytes, and found that periods of hypoxia resulted in elevations in apoptosis, p53 gene expression, and miR-449a levels. Downregulation of miR-449a with circular RNA circ-Ttc3 was able to decrease p53 expression and HaCaT apoptosis. They further found that by lowering miR-449a levels there was increased cellular viability by increasing the expression of PI3K/AKT, which has downregulated activity during in vivo human pressure sore development [28]. Increases in this PI3K/AKT activity by downregulating miR-449a may help protect from hypoxic injury and the development of pressure injury through improvements in cell viability.

Lowering miR-449a also increased NFκB expression levels. NFκB has pro-inflammatory activity as described above, with implications in the sustained pro-inflammatory immune response seen in pressure injury. NFκB, however, has both pro- and anti-apoptotic activity, and likely has a complex role in pressure ulcer pathogenesis and prevention [107,108,109,110]. Hypoxia increases NFκB expression in neutrophils and macrophages, which stimulates a pro-inflammatory response with increases in TNFα and NOS2, along with other pro-inflammatory cytokines [111,112]. Elevations in NFκB, TNFα, and NOS2 have been seen during the development of pressure ulcers in murine models, and are thought to contribute to the sustained pro-inflammatory immune response seen in these chronic wounds [110,112]. Conversely, hypoxia-induced NFκB activation reduces apoptosis and improves cell viability [113]. Further evaluation of the complex interactions of miR-449a and other miRNAs such as miR-21 on NFκB activity during pressure injury, and their balance in modulating apoptosis and the immune response, is still needed.

Niu et al. similarly showed that ligustrazine, an herbal ingredient extracted from the plant chuanxiong, has anti-inflammatory properties that improve pressure ulcer healing in a clinical setting [28]. To evaluate the effect of ligustrazine on hypoxia at the cellular level, the same group induced hypoxic injury in human umbilical vein endothelial cells (HUVECs). Hypoxia resulted in increased p53 and B-cell lymphoma (Bcl)-2 gene expression, with resultant increases in endothelial cell apoptosis and poor cell viability [83]. Hypoxia also resulted in the downregulation of miR-135b; however, treatment with ligustrazine was able to increase miR-135b levels, lower p53 expression, and reduce apoptosis. When miR-135b was inhibited, the effects of ligustrazine were lost, suggesting a direct role of miR-135b in the prevention of hypoxia-induced endothelial apoptosis. miR-135b was shown to increase the PI3K/AKT activity, the opposite effect of miR-449a, and acted to increase Jun N-terminal kinase/stress-activated protein kinase (JNK/SAPK) activity, suggesting multiple pathways in decreasing hypoxia-induced endothelial apoptosis [84]. miR-135b has been shown to have similarly protective effects in myocardial ischemia-reperfusion injury [114].

Local tissue ischemia and hypoxia in wounds increases pro-angiogenic signaling, but these pathways are often dysregulated in chronic wound models, leading to sustained hypoxic conditions. In murine pressure ulcers, there is a decrease in miR-200a activity, but miR-200a overexpression increases local angiogenesis [85]. miR-200a inhibits Kelch-like ECH-associated protein 1 (Keap-1), which allows for increased nuclear factor erythroid 2-related factor 2 (Nrf2) activity. Other miRNA that have similar activity on Nrf2 activity to promote angiogenesis and lower oxidative stress such as miR-23a and miR-29 have not been directly evaluated in pressure ulcers [85,115,116].

### 4.3. MiRNA Regulation of Oxidative Stress in Pressure Ulcers

Increased apoptotic debris and dysregulated clearance of toxic metabolites by lymphatic channels and immune cells result in elevated ROS that contribute to a sustained inflammatory response. Coupled with reperfusion injury, oxidative stress plays a critical role in the chronicity and poor wound healing associated with pressure wounds. When exposed to elevated levels of oxidative stress in a pressure ulcer model, keratinocytes have downregulated levels of miR-126, a regulatory microRNA involved in the Erk2 and PI3K/AKT pathways [86]. Overexpression of miR-126 resulted in increased PI3K/AKT activity, improved cell viability, and lower secondary ROS production. 

Dysregulation of miR-126 activity has been similarly shown in other chronic injury models. miR-126 overexpression enhances macrophage efferocytosis in diabetic cardiomyocytes, improving the transition to an anti-inflammatory state and improving injury repair [117]. miR-126 enhances angiogenesis and when negative pressure wound therapy is applied to diabetic wounds, the resultant elevation in miR-126 enhances Erk activity and correlated to increased capillary density within the wound bed [59,118,119]. Furthermore, in normal wound healing, miR-126 increases Erk2 and PI3K/AKT activity resulting in improved keratinocyte viability and migration, leading to improved wound closure [120].

MiR-145 is another microRNA implicated in reducing oxidative stress damage in models of pressure ulcers. Ge and Gao found that hydrogen peroxide-induced keratinocyte injury results in elevated ROS production and low miR-145 expression. When hydrogen peroxide-injured keratinocytes were treated with circular RNA ZNF609 silencer, there is an upregulation of miR-145. This increase in miR-145 resulted in lower ROS generation, decreased keratinocyte p53 and caspase 3 expression, and improved cell viability [87]. When miR-145 was silenced, there was a loss of anti-oxidant effects. miR-145 inhibits JNK and p38-MAPK signaling pathways, which are activated by ROS [87,121]. JNK has been shown to have a complex role in pressure ulcer pathophysiology, increasing ROS production but inhibiting apoptotic activity [84,87,121]. Increased levels of p38-MAPK have been shown to contribute to the ischemia-reperfusion injury seen in pressure ulcer pathogenesis, and miR-145 may downregulate this expression [23,25,87].

The complex mediation by miR-145 on inflammation, ROS, and apoptosis through JNK and p38-MAPK still requires further study in other pressure injury models. MiR-145 has been studied in a variety of neoplastic models, particularly melanoma, as a suppressor of tumor invasion and metastasis [122,123]. miR-145 is highly expressed in dermal fibroblasts, and increased expression of miR-145 has been shown to increase myofibroblast activity and wound contraction [124,125,126]. It is hypothesized that miR-145 is upregulated in pulmonary ischemia-reperfusion injury as a protective mechanism to mitigate the reperfusion injury [127]. Upregulation of miRNAs such as miR-126 and miR-145 could potentiate ROS production within chronic wounds like pressure ulcers. 

### 4.4. ECM Remodeling by miRNA-491-5p in Pressure Ulcers

Sustained ECM breakdown and impaired remodeling through overactivity of MMPs impairs wound closure in chronic wound models. Pressure ulcers have increased levels of MMP9 and decreased levels of tissue inhibitor metalloproteinase-1 (TIMP-1), and wounds with higher MMP9 and lower TIMP-1 activity heal slower than those with opposite relative levels [128]. Yang et al. found that patients who develop pressure ulcers following a hip fracture have elevated levels of MMP9 activity. Patients who have a Rs1056629 polymorphism in the MMP9 gene are more likely to develop pressure ulcers due to impaired miR-491-5p binding, which typically inhibits MMP9 activity [88].

MiR-491 has effects on MMP activity depending on the tissue type. The inhibition of MMP9 by miR-491-5p has been implicated in lowering metastasis and tissue invasion in hepatocellular carcinoma and osteosarcoma [129,130]. MiR-491-5p also decreases cellular proliferation and motility in neoplastic models through activity on other gene targets, but further investigation of its effects on MMP activity during impaired wound healing is required [131,132]. Within placental tissue, miR-491-5p inhibits MMP9 activity, leading to lower trophoblast invasion and higher rates of pre-eclampsia [133]. MMP activity contributes to chronic wound formation not only through degradation of the extracellular matrix, but also through increased liberation of inflammatory cytokines such as TNFα and IL-1β, so the activity of miR-491-5p in lowering pressure ulcer risk may involve both lowering inflammatory signaling and decreasing ECM breakdown and the regulation of MMP9 activity [134,135].

### 4.5. Targeting miRNAs to Enhance Pressure Ulcer Healing

The regulation of miRNA expression levels within pressure ulcers has not been widely studied in clinical models, but offers a potential avenue for improved prevention and therapeutic strategies. In preclinical studies of hypoxia-induced endothelial cell injury, ligustrazine was shown to reduce hypoxic injury through upregulation of miR-135b, and in a clinical setting ligustrazine accelerates pressure ulcer healing; however, whether this therapeutic acts through regulation of miR-135b in a clinical setting has not been described [28,83]. MicroRNAs can also be delivered directly to skin and soft tissue to enhance wound healing such as through viral vectors, lipid vesicles, or nanoparticles, although this has largely been demonstrated in other wound models [76,136,137]. MiR-126, which decreases oxidative damage in preclinical models of pressure injury, has been successfully delivered via liposomal conjugation to improve angiogenesis and wound healing following hindlimb ischemia [138]. 

## 5. Conclusions

Pressure injury has a significant physical, psychosocial, and economic burden on patient care. MicroRNAs have been described in the pathogenesis and treatment of chronic wounds, but their specific roles in the pathogenesis of pressure ulcers have not been reviewed previously. While correlations with other chronic wound models can be made to improve our understanding of pressure sore healing, the interplay of ischemia-reperfusion injury, impaired lymphatic drainage, and cellular deformation following mechanical pressure has unique pathophysiologic mechanisms. Critical pathways involved in miRNA regulation of pressure ulcer pathophysiology include PI3K/AKT, p38-MAPK, NFκB, and MMP9. Prevention of pressure injury remains of paramount importance, but early therapeutic intervention to prevent stage progression and chronic wound formation will be needed. MiRNAs offer potential regulatory targets to prevent the ongoing pressure injury and improve pressure ulcer resolution. 

## Figures and Tables

**Figure 1 ijms-22-00064-f001:**
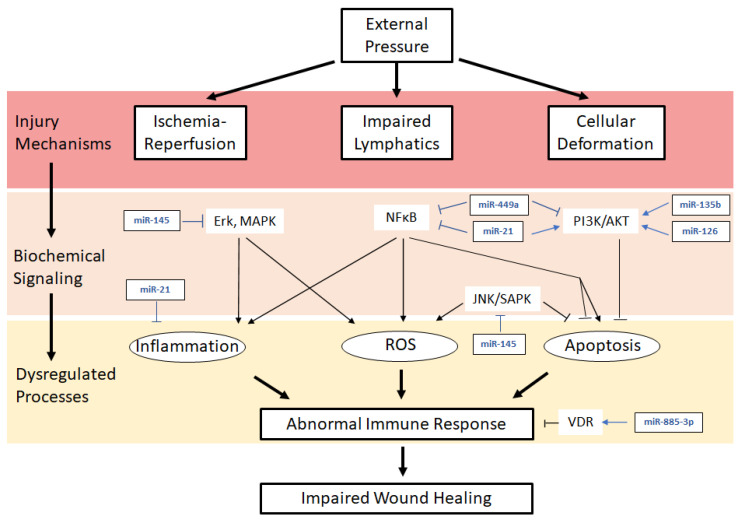
Pathophysiologic mechanism of pressure ulcers and current understanding of miRNA regulation. Ischemia-reperfusion, lymphatic channel obstruction, and cellular deformation result in elevated inflammation, ROS, and apoptosis, which contribute to immune response dysregulation and impaired wound healing. Flat head arrows indicate inhibition, pointed arrows indicate activation.

**Table 1 ijms-22-00064-t001:** Summary of known microRNAs implicated in pressure ulcer pathogenesis.

Study	MicroRNA	Effect
Song et al. 2019 [81]	miR-21	Levels are decreased in inflammatory conditions. Decreases pro-inflammatory iNOS, COX-2, IL-6, and IL-1β to improve keratinocyte viability
Ji et al. 2017 [82]	miR-885-3p	Increases vitamin D receptor (VDR) activity and lowers pressure ulcer risk. Patients with Rs739836 VDR polymorphism have poor miR-885-3p binding and increased risk of ulcer development
Wei et al. 2019 [83]	miR-135b	Decreases hypoxic injury by increasing PI3K/AKT and increasing JAK/SAPK activity, lowering endothelial cell apoptosis
Yu et al. 2020 [84]	miR-449a	Increases hypoxic injury by inhibiting PI3K/AKT and NFκB activity, increasing keratinocyte apoptosis
Chen et al. 2019 [85]	miR-200a	Increases angiogenesis in murine model of pressure ulcers by decreasing Keap1, an upstream inhibitor of Nrf2
Zhang et al. 2019 [86]	miR-126	Decreases oxidative damage in keratinocytes by reducing ROS. Increases PI3K/AKT/mTOR activity
Ge and Gao 2020 [87]	miR-145	Decreases oxidative damage in keratinocytes by reducing ROS. Decreases JNK and p38-MAPK activity lowering ROS
Yang et al. 2018 [88]	miR-491-5p	Impaired binding to the MMP9 gene results in an increased risk of pressure ulcer development

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
