# Peer review of "Role of microRNAs in Pressure Ulcer Immune Response, Pathogenesis, and Treatment"

_ijms, 2020, doi:10.3390/ijms22010064_

Round 1

Reviewer 1 Report

The manuscript by Niemiec and colleagues concisely elaborates on the pathophysiology of pressure ulcers and elucidated ischemia reperfusion, impaired lymphatic drainage and cellular deformation as the key events leading to the development of these ulcers. They went further to describe the biological processes involved namely inflammation, apoptosis und reactive oxygen species which culminate in deregulating immune responses. 

MicroRNAs are later introduced as important mediators in the dysregulation of the above-mentioned processes and thus inherently serve as treatment targets.

The manuscript is well put together und despite the thin literature and body of work on miRNAs and pressure ulcers, the authors have managed to give significant evidence of the role these molecules play in the mediation of these ulcers.

The following point(s) need to be addressed before the manuscript can be considered for publication:

Despite being involved in the pathogenesis of pressure ulcers, it is still not sufficiently clear from the manuscript how miRNAs can be used as agents of healing as the subtitle “4” suggests. It will be nice if the authors could add a small paragraph explaining the prospects and strategies that could be employed to achieve this.

Author Response

Thank you for your thorough review and comments. An additional section 4.5. has been added on lines 353 to 364 with examples of how miRNA regulation has been used in pressure ulcer therapy, along with references to miRNA delivery mechanisms used in other chronic wound models.

Reviewer 2 Report

Within their manuscript authors present data regarding the effect of zebra fish embryo extract on human ASC in late passages. Therefore they evaluated cell viability as well as expression of several g Within their manuscript authors describe pressure ulcers. This review focuses on key biochemical pathways in the pathogenesis of pressure injury and how this signaling leads to impaired wound healing. I suggest some changes below. I think the manuscript will benefit from them.

Minor comments:

  1. The quality of Figure 1 is not sufficient. Please improve it.
  2. Conclusions: “To our knowledge, this review is the first to summarize current understanding of the role of miRNAs in the pathogenesis and management of pressure injury. Pressure injury has a significant physical, psychosocial, and economic burden on patient care. Pressure injury…“                                                                                                                                                             To my knowledge, this is not the first review describing the role of miRNAs in the pathogenesis and management of chronic wound. Therefore, please chamge this statement and add some more references. Second, there is "pressue injury" in every sentence.

Author Response

  1. Thank you for your thorough review and comments. Figure 1 has been revised to simplify the signaling processes. An improved description of the figure legend has been included as well.
  2. Within the conclusion (manuscript lines 366-379), the wording has been revised to reduce repetitive language. Additionally, the emphasis of describing microRNA pathophysiology specific to pressure ulcers rather than broadly in chronic wounds has been clarified. This point has been introduced in lines 48-49 as well.